# Açai Berry Administration Promotes Wound Healing through Wnt/β-Catenin Pathway

**DOI:** 10.3390/ijms24010834

**Published:** 2023-01-03

**Authors:** Livia Interdonato, Ylenia Marino, Gianluca Antonio Franco, Alessia Arangia, Ramona D’Amico, Rosalba Siracusa, Marika Cordaro, Daniela Impellizzeri, Roberta Fusco, Salvatore Cuzzocrea, Rosanna Di Paola

**Affiliations:** 1Department of Chemical, Biological, Pharmaceutical and Environmental Sciences, University of Messina, Viale Ferdinando Stagno D’Alcontres 31, 98166 Messina, Italy; 2Department of Biomedical, Dental and Morphological and Functional Imaging, University of Messina, Via Consolare Valeria, 98125 Messina, Italy; 3Department of Veterinary Sciences, University of Messina, 98168 Messina, Italy

**Keywords:** wound healing, Wnt, açai berry, inflammation

## Abstract

Recently, wound healing has received increased attention from both a scientific and clinical point of view. It is characterized by an organized series of processes: angiogenesis, cell migration and proliferation, extracellular matrix production, and remodeling. Many of these processes are controlled by the Wnt pathway, which activates them. The aim of the study was to evaluate the molecular mechanism of açai berry administration in a mouse model of wound healing. CD1 male mice were used in this research. Two full-thickness excisional wounds (5 mm) were performed with a sterile biopsy punch on the dorsum to create two circular, full-thickness skin wounds on either side of the median line on the dorsum. Açai berry was administered by oral administration (500 mg/kg dissolved in saline) for 6 days after induction of the wound. Our study demonstrated that açai berry can modulate the Wnt pathway, reducing the expression of Wnt3a, the cysteine-rich domain of frizzled (FZ)8, and the accumulation of cytosolic and nuclear β-catenin. Moreover, açai berry reduced the levels of TNF-α and IL-18, which are target genes strictly downstream of the Wnt/β-catenin pathway. It also showed important anti-inflammatory activities by reducing the activation of the NF-κB pathway. Furthermore, Wnt can modulate the activity of growth factors, such as TGF-β, and VEGF, which are the basis of the wound-healing process. In conclusion, we can confirm that açai berry can modulate the activity of the Wnt/β-catenin pathway, as it is involved in the inflammatory process and in the activity of the growth factor implicated in wound healing.

## 1. Introduction

Wound healing (WH) is a vital physiological process that ensures the skin’s integrity after injuries. It is caused by accidents, external influences, medical operations, or other circumstances. Wound healing is a complex process that involves numerous steps in both humans and animals, including inflammation, proliferation, and remodeling [1]. The inflammatory phase is characterized by hemostasis and inflammation. It includes cellular processes, such as the invasion of leukocytes and cytokine production, which starts the proliferative response for wound repair, as well as vascular reactions characterized by blood coagulation and hemostasis [2]. The main pathway involved in the healing process is the Wnt signaling pathway. The Wnt pathway’s activation regulates a multitude of events involved in adult homeostasis and embryonic development. Wnt pathways include noncanonical β-catenin–independent signaling pathways and canonical β-catenin signaling pathways [3]. Wnt family proteins bind to cell surface receptors that are expressed by the frizzled (Fz) gene family. When the destruction complex is inactive, β-catenin accumulates in the cytosol, which then can enter the nucleus and bind to Tcf, the transcriptional effector of the Wnt pathway [4]. By lowering paracellular permeability, activation of the Wnt pathway also strengthens the barrier function of cultured cells. A Wnt ligand receptor called FZD7 limits paracellular permeability and interacts physically with complex elements of the adherens junction. These mechanistic discoveries support the idea that the Wnt pathway is crucial for maintaining vascular homeostasis [5]. The sustained activation of the β-catenin-dependent pathway causes the development of epithelial appendages, such as epithelial cysts inside the dermis, hair follicles, and sporadic sebocytes. In the wound bed, primarily in the regenerating epithelium, Wnt3a activates the endogenous Wnt pathway. The extent to which L-Wnt3a enters the tissues is directly related to the penetration of the epithelium. Epithelial cells are impacted by the interaction between the transforming growth factor-beta (TGF-β) and Wnt/β-catenin signaling. The expression of Wnt3a, a well-known β-catenin signal inducer, was shown to be elevated in patients with active cutaneous keloids, and it has been linked to the endothelium-to-mesenchymal transition [6]. The epidermis’s overexpression of Wnt genes causes dermal fibroblast growth [7]. In particular, the Wnt pathway is involved in the modulation of the activity of growth factors. Known as a Wnt target, VEGF has β-catenin response elements in its promoter. It has been discovered that a T-cell factor-4 (TCF-4) binding region in the VEGF promoter, located upstream of the transcriptional start site, is a key mediator of this action [8]. This growth factor specifically targets vascular endothelial cells, inducing proliferation and migration. Indeed, it drives angiogenesis and increases vascular permeability [9]. Another growth factor involved in the wound-healing process is TGF-β. When granulation tissue is forming, TGF-β substantially encourages fibroblast and endothelial cell migration as well as fibroblast deposition of extracellular matrices [2]. Since these processes involve TGF-β signaling, medicinal drugs that target the TGF-β pathway may result in better wound healing and scarring [10]. TGF-β stimulation boosts the activity of responsive elements in reporter tests and causes nuclear β-catenin accumulation in cultured fibroblasts. Additionally, adenoviral overexpression of a constitutively active TGF-β receptor type I caused fibrosis, and the nuclear accumulation of β-catenin in fibroblasts triggered the canonical Wnt pathway [11]. Furthermore, the dysregulation of various proinflammatory factors, including IL-1 beta, IL-6, and TNF-α, are increased in chronic wounds. This causes excessive disintegration of the local extracellular matrix and impairs cell migration. Crosstalk between NF-κB signaling and canonical Wnt/β-catenin signaling during inflammation has been extensively studied. Studies have demonstrated modulation of inflammatory responses via the interaction between Wnt/β-catenin signaling and NF-κB. Proinflammatory cytokines, including IL-6, IL-1β, and TNF-α, were augmented [12].

Açai berries have been described to have a range of health benefits and can be a potential treatment. This fruit, which has a sour and delicious flavor, is made from an Amazon-only palm species called Euterpe oleracea. In particular, it contains antioxidants, unsaturated fats (omega-3, omega-6, and omega-9), vitamins, and minerals that give the fruit its hypocholesterolemic [13], immunostimulant [14], anti-inflammatory, and antioxidant qualities [15]. This food has drawn the attention of researchers due to its antioxidant effects [16].

## 2. Results

### 2.1. Effect of Açai Berry on Tissue Repair

To observe the effect of açai berry on changes in dermal tissue induced by WH, H&E staining was performed (Figure 1). H&E staining showed extensive tissue damage in the WH group (Figure 1C,C-1,C-2,E), as compared to the tissue harvested from the sham group (Figure 1A,A-1,A-2,E) and sham + açai berry group (Figure 1B,B-1,B-2). The oral administration of açai berry (Figure 1D,D-1,D-2,E) caused improvements in wound closure, granulation, and re-epithelialization, all of which contributed to a nearly complete restoration of the epidermis in this group. Toluidine blue staining was employed (Figure 2) to observe the distribution of mast cells. We detected a higher number of mast cells in the WH group (Figure 2B,B-1,D), as compared to the sham group (Figure 2A,A-1,D). Açai berry administration significantly reduced mast cell recruitment (Figure 2C,C-1,D).

### 2.2. Açai Berry Effect on Fibrosis Condition

The development of wound lesions is associated with a high degree of tissue fibrosis and increased neovascularization events. The degree of fibrosis was evaluated by using Masson’s trichrome staining. This staining showed a clear, visible, fine, and coarse collagen deposition and its arrangement in the wounded skin. The WH group (Figure 3B,B-1) showed a significant reduction in collagen fibers on the sixth day after wound induction; meanwhile, the açai berry group showed a stimulating action on collagen arrangement (Figure 3C,C-1). Immunohistochemical analysis was employed to evaluate the expression of VEGF (Figure 3D–F), a key regulator of angiogenesis. Instead, TGF-β (Figure 3H–J) is involved in the homeostasis of epithelial and endothelial tissues, hematopoietic and immune systems, and skeletal and neural organs. In particular, we showed an increase in VEGF expression in the WH group (Figure 3E) as compared to the sham group (Figure 3D); however, açai berry can remodulate VEGF levels (Figure 3F). Even TGF-β expression can be modulated by açai berry. In fact, the WH group (Figure 3I), as compared to the sham group (Figure 3H), showed higher levels; the administration of açai berry was able to re-establish normal levels of TGF-β (Figure 3J).

### 2.3. Effect of Acai on ICAM-1 and P-Selectin Expression

To evaluate whether the treatment with açai berry is capable of modulating adhesion molecules, we performed immunohistochemical staining for ICAM-1 and P-selectin. The results showed an increase in the expression of these proteins in the WH groups (Figure 4B,F) compared to the sham animals (Figure 4A,E); however, the açai berry treatment (Figure 4C,G) is capable of modulating this expression.

### 2.4. Effect of Acai on Nitrotyrosine and Parp-1 Expression

Immunohistochemical analysis was performed to evaluate the expression of nitrotyrosine and Parp-1.

An overexpression of these biomarkers was shown in the WH group (Figure 5B), as compared to the sham group (Figure 5A); the administration of açai berry was able to reduce this increase (Figure 5C). PARP-1 is involved in normal or abnormal recovery from DNA damage [17]. The WH group (Figure 5F), compared to the sham group (Figure 5E), showed higher levels of Parp-1; thus, the administration of açai berry was able to re-establish normal levels of these proteins (Figure 5G).

### 2.5. Effect of Açai Berry on Inflammation

To evaluate the anti-inflammatory activity of açai berry, we analyzed the levels of many inflammatory cytokines.

IL-1β (Figure 6A), TNF-α (Figure 6B), and IL-6 (Figure 6C) levels were significantly increased in the WH group compared to sham mice. On the contrary, cytokine release was markedly reduced in mice treated with açai berry. Another inflammatory pathway was represented by NF-κB; Western blot analysis showed reduced IκB-α (Figure 6D) expression in the cytosolic fraction of the samples harvested from the WH group, as compared to the sham group. Açai berry administration increased cytosolic IκB-α expression and restored it to a basal level. In the nuclear fraction, NF-κB (Figure 6E) expression was increased in the WH group, as compared to the sham mice. The açai berry treatment significantly reduced nuclear NF-κB expression.

### 2.6. Effect of Açai Berry Administration on Wnt/FZ/β-Catenin Pathway

To analyze the effect of açai berry on the Wnt pathway, we performed a Western blot analysis (Figure 7). This analysis showed an increased expression of Wnt3a and FZ8 in the WH group, as compared to the sham group. The administration of açai berry was able to reduce this increase. Additionally, we determined the expression of β-catenin, which is a downstream effector of the Wnt/β-catenin pathway. Our analysis showed an important increase in both the cytosolic and nuclear fractions of β-catenin expression. Açai berry, at a dose of 500 mg/kg, can reduce β-catenin expression in the cytoplasm and nucleus.

## 3. Discussion

The integrity of healthy skin plays a crucial role in maintaining the physiological homeostasis of the human body since the skin regulates temperature and serves as a barrier against dangerous microorganisms [18]. Skin injuries are highly frequent ailments that can happen for many reasons. Hemostasis, inflammation, proliferation, and tissue remodeling or resolution make up the four integrated and overlapping physiological events that are a part of the complicated physiological process of wound healing [1]. Wound healing begins with inflammation and exudate secretion. After that, inflammation is suppressed, leading to re-epithelization and wound contraction [19]. Hemostasis, chemotaxis, and enhanced vascular permeability define the inflammatory phase, which limits additional damage, closes the wound, removes cellular debris and microorganisms, and promotes cellular migration. Typically, the inflammatory stage lasts several days [20]. The main pathway involved in the healing process is the Wnt signaling pathway. In recent times, several studies have been conducted on açai berries to evaluate their anti-inflammatory activity and antioxidant capacity [21]. In this study, we analyzed the effects of açai berry on the Wnt pathway in a mouse model of wound healing. Our first results were from a histological analysis, in which we evaluated the tissue damage with H&E staining. A pattern of leucocytic infiltration into the wound emerges when the recruitment of leukocytes from the circulation increases, which is similar to what happens in other acute inflammatory diseases [22]. Our analysis showed extensive tissue damage in the WH group as compared to the sham group, whereas mice treated with açai berry showed a reduction in their inflammation condition. Inflammation in the wound is generally caused by resident mast cells (MCs) and precursors of MCs are recruited from the circulation, along with monocytes, neutrophils, and T cells that enter the tissue from the blood after damage [22]. To confirm the activity of MCs, we performed a second analysis. Toluidine blue staining showed the distribution of mast cells. The treatment with açai berry significantly reduced the presence of these immune cells and their infiltration into wound beds as compared to the WH group. The development of wound lesions is associated with a high degree of tissue fibrosis [23]; for this reason, we performed Masson’s trichrome staining to evaluate the fibrosis condition. Our results showed a significant reduction in collagen fibers in the WH group as compared to the sham group, whereas the oral treatment with açai berry showed a stimulating action on collagen arrangement. Wound healing is associated with endothelial cell adhesion molecules (CAMs) since CAMs influence the temporal and lineage profiles of extravasated leukocytes within a wound [24]; an immunohistochemical analysis showed that açai berry modulated the expression of adhesion molecules. In particular, we analyzed ICAM-1 and P-selectin levels and our results showed higher levels of these proteins in the WH group and lower levels in the group treated with açai berry. ROS are centrally involved in all wound-healing processes. Excessive production of ROS or impaired ROS detoxification causes oxidative damage [25]. Nitrotyrosine is a biomarker of oxidative stress formed due to the nitration of protein-bound and free tyrosine residues by reactive peroxynitrite molecules [26]. The immunohistochemical analysis of nitrotyrosine expression showed an increase in its level in the WH group as compared to the sham group, whereas the açai berry treatment was able to restore the normal levels of this biomarker. Another immunohistochemical analysis was performed to evaluate the activity of Parp; it is involved in recovery from DNA damage [27]. In the WH group, there was an overexpression of this enzyme, whereas the açai berry group demonstrated a normal level of expression. TGF-β is a family of growth factors implicated in a number of fundamental cellular functions. In particular, TGF-β is involved in all stages of WH: inflammation, angiogenesis, fibroblast proliferation, collagen synthesis, and deposition, as well as restoration of the new extracellular matrix [28]. VEGF intervenes in the angiogenic activity during the proliferative stage of WH [29]. VEGF likely encourages collagen deposition and epithelialization in addition to promoting angiogenesis, which stimulates wound healing [30]. The activity of growth factors, such as TGF-β and VEGF, is regulated by the Wnt family [31]; for this reason, we explored the activation of the Wnt/β-catenin signaling pathway. It can increase the VEGF level of expression in the WH group, whereas açai berry oral administration can modulate the Wnt activity and VEGF expression. Furthermore, TGF-β expression showed lower levels in the WH group as compared to the sham group, and the administration of açai berry was able to play an inhibitory role in Wnt/β-catenin signaling and remodulate the TGF-β levels. TGF-β’s capacity to stimulate angiogenesis might be associated with its ability to stimulate VEGF expression at the site of the wound. The inflammatory process, associated with wound healing, involves an increase in cytokine levels, such as IL-6, IL-1β and TNF-α; the Wnt/β-catenin pathway is involved in inflammatory cytokine expression [3]. In particular, our analysis showed higher levels of expression of these cytokines, whereas the açai berry group showed anti-inflammatory activity, reducing their levels and modulating Wnt activity in these cytokines. Cross-regulation between the Wnt and nuclear factor NF-κB signaling pathways has emerged as an important factor for the regulation of a diverse array of genes and pathways active in the inflammation state [32]; for this reason, we performed a Western blot analysis to evaluate the activity of açai berry and its anti-inflammatory activity in the NF-κB pathway. In particular, we demonstrated that açai berry reduces IκB-α cytosolic degradation and NF-κB nuclear expression, downregulating the inflammation associated with wound healing. To confirm the effects of açai berry on the Wnt/β-catenin pathway, we performed Western blot analyses on Wnt3a, FZ8, and β-catenin expressions. Western blot analysis revealed that the Wnt3a and FZ8 expression was higher in the WH group as compared to the sham group. However, taking açai berry supplements helped to lessen this increase. β-Catenin regulates fibroblast behavior during the proliferative phase of cutaneous wound healing. β-Catenin protein levels and the transcriptional activity are enhanced in mouse dermal fibroblasts during the proliferative phase of healing and return to baseline during the remodeling phase [33].The expression of β-catenin, a downstream effector of the Wnt/β-catenin pathway, was also measured via Western blot analysis. Our study revealed a significant rise in β-catenin expression in both the cytosolic and nuclear fractions. At a dosage of 500 mg/kg, açai berries can lessen the expression of β-catenin in the cytoplasm and nucleus, remodeling the levels of expression of this enzyme.

## 4. Materials and Methods

### 4.1. Animals

Male CD1 mice (25–30 g, Envigo, Milan, Italy) were housed in a controlled environment with food and water ad libitum. The University of Messina Review Board for Animal Care (OPBA) approved the study. All in vivo experiments followed the new directives of the USA, Europe, and Italy, as well as the ARRIVE guidelines.

### 4.2. Induction of Experimental Wound Healing

Mice were anesthetized with isoflurane prior to shaving the hair on their back areas, and then, a betadine scrub and 70% ethanol were applied alternately 3 times to prepare the dorsum for wounding. Two full-thickness excisional wounds (5 mm) were generated with a sterile biopsy punch (KAI Corporation, Tokyo, Japan) on the dorsum to create two circular, full-thickness skin wounds on either side of the median line on the dorsum. Mice were sacrificed after 7 days and the wounds were separated into two segments. The first one was used for histology and the second one for molecular analysis.

### 4.3. Experimental Groups

The animals were randomly distributed into the following groups (N = 12):-WH group: mice were subjected to full-thickness excisional wounds as described above.-WH + açai berry group: mice received oral administration of açai berry (500 mg/kg dissolved in saline) for 6 days after induction of the wound.-Sham group: mice were subjected to all procedures described above, except that the full-thickness excisional wounds were not applied and saline was administered for 6 days after.-Sham + açai berry group: mice were subjected to all procedures described above, except that the full-thickness excisional wounds were not applied and saline with açai berry was administered (500 mg/kg dissolved in saline) for 6 days after.

### 4.4. Histological Analysis

Wound tissue specimens were quickly removed and were fixed in 10% formalin for at least 24 h at room temperature. After dehydration with ethanol and fixing with paraffin, 7 μm sections were prepared and, subsequently, stained with H&E and with toluidine blue to determine mast cell (MC) degranulation [34]. Every section was examined using a Leica DM6 microscope (Leica Microsystems S.p.A., Milan, Italy) in association with the Leica LAS X Navigator software (Leica Microsystems S.p.A., Milan, Italy) [35].

### 4.5. Immunohistochemical Localization of VEGF, ICAM-1, P-Selectin, Nitro, and Par

The immunohistochemical evaluation for vascular endothelial growth factor (VEGF), intercellular adhesion molecule 1 (ICAM-1), and P-selectin was realized as previously described [36]. Slices were incubated overnight: anti-VEGF mouse monoclonal antibody (Santa Cruz Biotechnology, Heidelberg, Germany; 1:100 in PBS, *v*/*v*), anti-ICAM-1 mouse monoclonal antibody (Santa Cruz Biotechnology; 1:100 in PBS, *v*/*v*), and anti-P-selectin mouse monoclonal antibody (Santa Cruz Biotechnology; 1:100 in PBS, *v*/*v*). The samples were washed with PBS and incubated with secondary antibodies. Specific labeling was identified with a biotin-conjugated goat anti-rabbit IgG and avidin–biotin peroxidase complex (Vector Laboratories, Burlingame, CA, USA) [37]. The stained sections were observed using a Leica DM6 microscope (Leica Microsystems S.p.A., Milan, Italy), following a typical procedure [38].

### 4.6. Western Blot

Western blot analysis was realized as previously described [18]; anti-IκB-α (1:1000, SCB, #sc1643), anti-NF-κB p65 (1:1000; SCB, #sc8414) anti-Wnt3a (Santa Cruz Biotechnology (SCB), Dallas, TX, USA, sc-80457), anti-FZ8 (Bioworld Technology, St. Louis Park, MN, USA). They were also incubated with a β-actin antibody (1:5000; Santa Cruz Biotechnology) for cytosolic proteins or β-catenin antibody (1:5000; Santa Cruz Biotechnology) for nuclear proteins. Protein expression was quantified via densitometry with Bio-Rad ChemiDocTM XRS+ software and normalized with the housekeeping genes β-actin and lamin A/C, as previously reported [39].

### 4.7. Elisa

The expression of interleukin (IL)-18, tumor necrosis factor (TNF)-α and IL-1β was measured using ELISA kits (R&D Systems, Minneapolis, MN, USA), following the manufacturer’s instructions [40].

### 4.8. Materials

Unless otherwise stated, all compounds used in this study were purchased from Sigma-Aldrich Company Ltd. (Milan, Italy). Freeze-dried açai extract was dissolved in saline; however, previous studies showed that it could be also dissolved in water. It was administered orally at a dose of 500 mg/kg. This substance (Cas Number: 879496-95, 906351-38-0) was purchased from Farmalabor Srl (Canosa di Puglia, Barletta, Italy). The dose and the route of administration of açai berry were chosen based on our previous study, where we performed a dose–response experiment [16,41]. No significant difference was found between the sham and sham + açai berry groups; therefore, only data regarding the sham groups are shown.

### 4.9. Statistical Evaluation

All values are expressed as means ± standard errors of the means (SEM) of N observations. The images shown are representative of at least 3 experiments performed on tissue sections collected from all animals in each group, which were carried out on diverse experimental days. For in vivo studies, N represents the number of animals used. The results were analyzed by using a one-way ANOVA, followed by a Bonferroni post hoc test for multiple comparisons. A *p*-value less than 0.05 was considered significant. * *p* < 0.05 vs. sham; # *p* < 0.05 vs. WH; ** *p* < 0.01 vs. sham; ## *p* < 0.01 vs. WH; *** *p* < 0.001 vs. sham; ### *p* < 0.001 vs. WH.

## 5. Conclusions

In this study, we demonstrated that açai berry was able to modulate the activation of the Wnt/β-catenin signaling pathway in a mouse model of wound healing. In particular, the oral administration of açai berry was able to improve tissue damage, reduce cell infiltration, and improve tissue quality. Additionally, it reduced the levels of TNF-α and IL-18 that are strictly downstream of the Wnt/β-catenin pathway and modulated the activity of growth factors, such as TGF-β and VEGF, which are the basis of the wound-healing process. Açai berry also showed important anti-inflammatory and antioxidant activities, reducing the activation of the NF-κB pathway and the expression of nitrotyrosine and PARP-1.

## Figures and Tables

**Figure 1 ijms-24-00834-f001:**
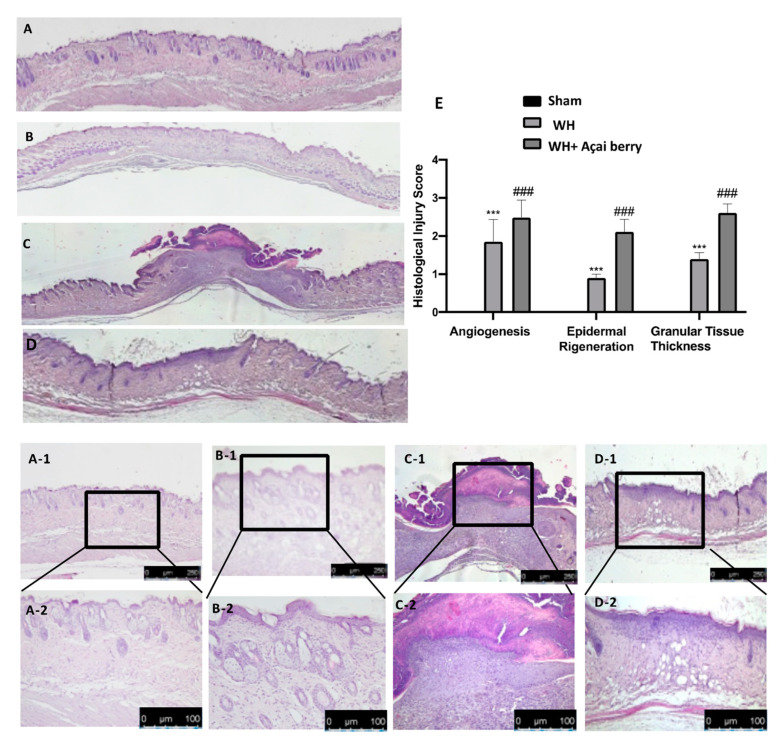
Histological analysis: sham (**A**,**A-1**,**A-2**), sham + açai Berry (**B**,**B-1**,**B-2**), WH (**C**,**C-1**,**C-2**), WH + açai berry (**D**,**D-1**,**D-2**), histological injury score graph (**E**). A 10× and 20× magnification is shown. A demonstrative blot of lysates with a densitometric analysis for all animals is shown. Data are expressed as the mean ± SEM of N = 6 mice/group. *** *p* < 0.001 vs. sham; ### *p* < 0.001 vs. WH.

**Figure 2 ijms-24-00834-f002:**
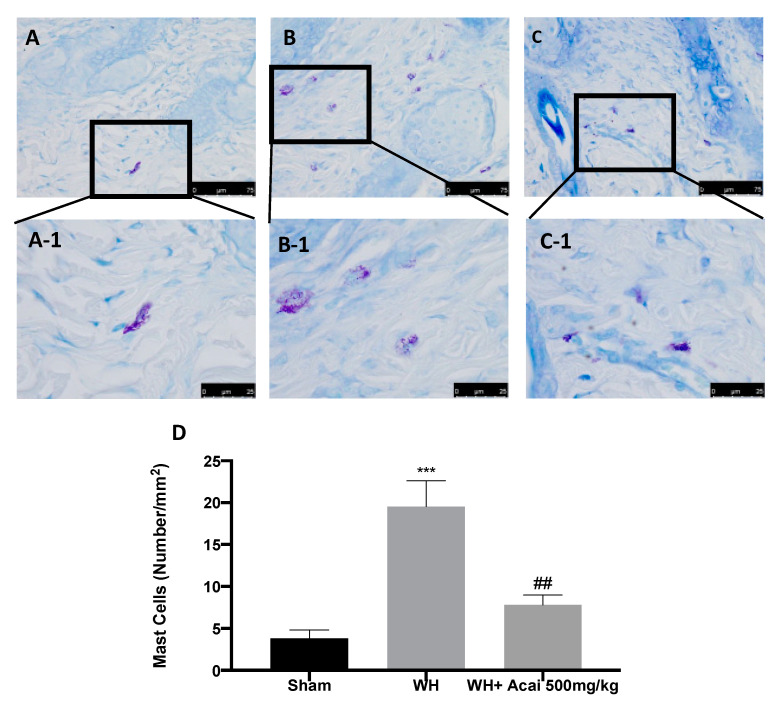
Mast cells indicated by toluidine blue staining: sham (**A**,**A-1**), WH (**B**,**B-1**), WH + açai berry (**C**,**C-1**), mast cell count (**D**). A 75× magnification is shown (25 µm scale bar). A demonstrative blot of lysates with a densitometric analysis for all animals is shown. Data are expressed as the mean ± SEM of N = 6 mice/group. *** *p* < 0.001 vs. sham; ## *p* < 0.01 vs. WH.

**Figure 3 ijms-24-00834-f003:**
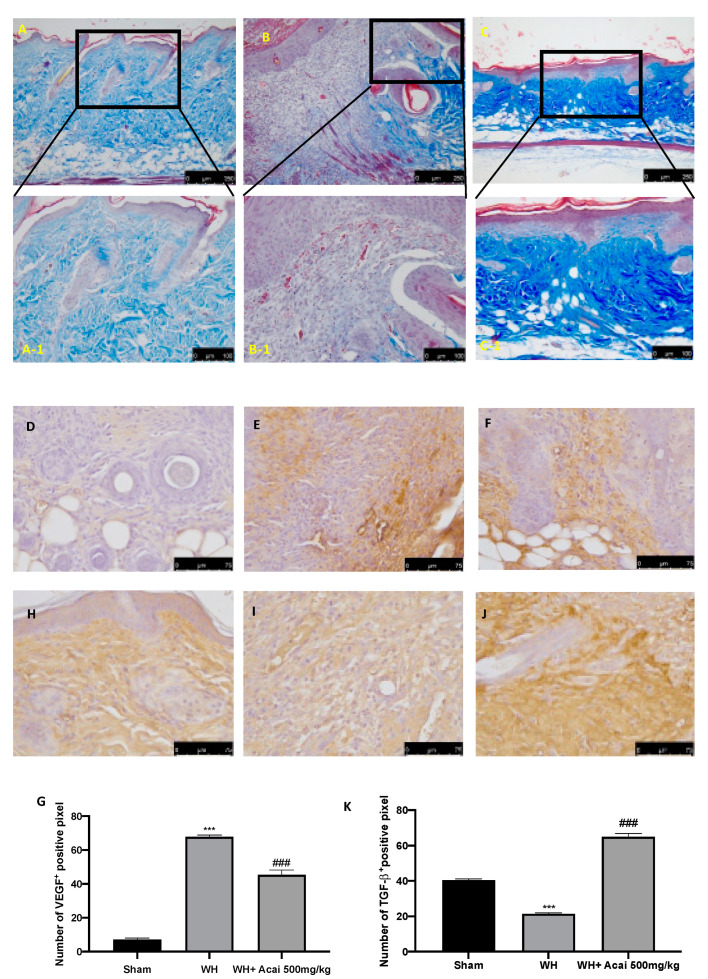
Masson’s trichrome staining to observe changes in dermal collagen through optical microscopy; sham (**A**,**A-1**), WH (**B**,**B-1**), WH + açai berry (**C**,**C-1**). A 10× and 20× magnification is shown. Immunohistochemical analysis of VEGF (**G**) and TGF-β (**K**); sham (**D**,**H**), WH (**E**,**I**), WH + açai berry (**F**,**J**). A 75× magnification is shown. A demonstrative blot of lysates with a densitometric analysis for all animals is shown. Data are expressed as the mean ± SEM of N = 6 mice/group. *** *p* < 0.001 vs. sham; ### *p* < 0.001 vs. WH.

**Figure 4 ijms-24-00834-f004:**
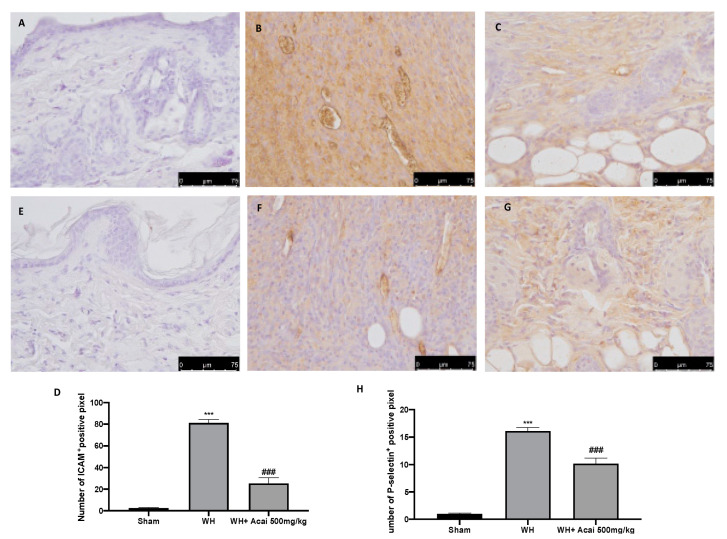
Immunohistochemical analysis of adhesion molecules. ICAM expression: sham (**A**), WH (**B**), WH + açai berry (**C**), % positive pixel for ICAM (**D**). P-selectin expression: sham (**E**), WH (**F**), WH + açai berry (**G**), % positive pixel for P-selectin (**H**). A 75× magnification is shown. A demonstrative blot of lysates with a densitometric analysis for all animals is shown. Data are expressed as the mean ± SEM of N = 6 mice/group. *** *p* < 0.001 vs. sham; ### *p* < 0.001 vs. WH.

**Figure 5 ijms-24-00834-f005:**
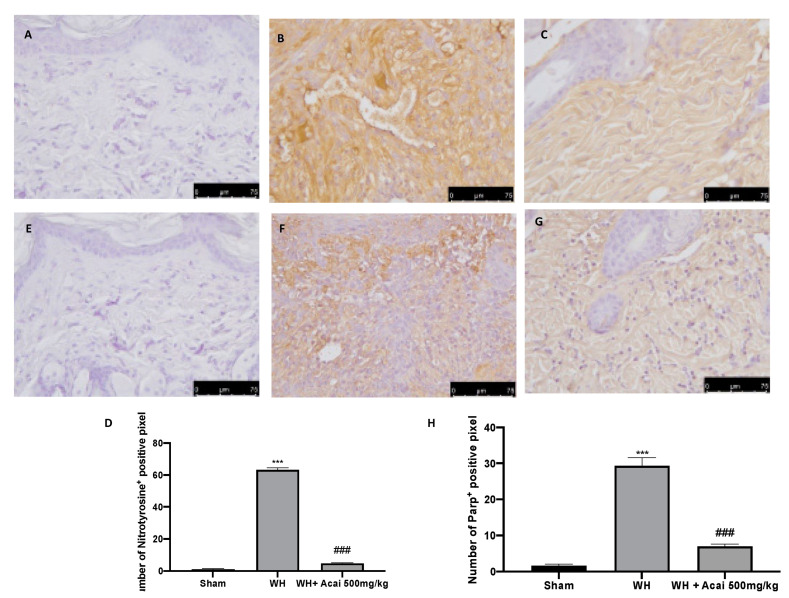
Immunohistochemical analysis was performed to evaluate the expressions of biomarkers of oxidative stress. nitrotyrosine expression: sham (**A**), WH (**B**), WH + açai berry (**C**), % positive pixel for nitrotyrosine (**D**). Parp-1 expression: sham (**E**), WH (**F**), WH + açai berry (**G**), % positive pixel for Parp-1 (**H**). A 75× magnification is shown. A demonstrative blot of lysates with a densitometric analysis for all animals is shown. Data are expressed as the mean ± SEM of N = 6 mice/group. *** *p* < 0.001 vs. sham; ### *p* < 0.001 vs. WH.

**Figure 6 ijms-24-00834-f006:**
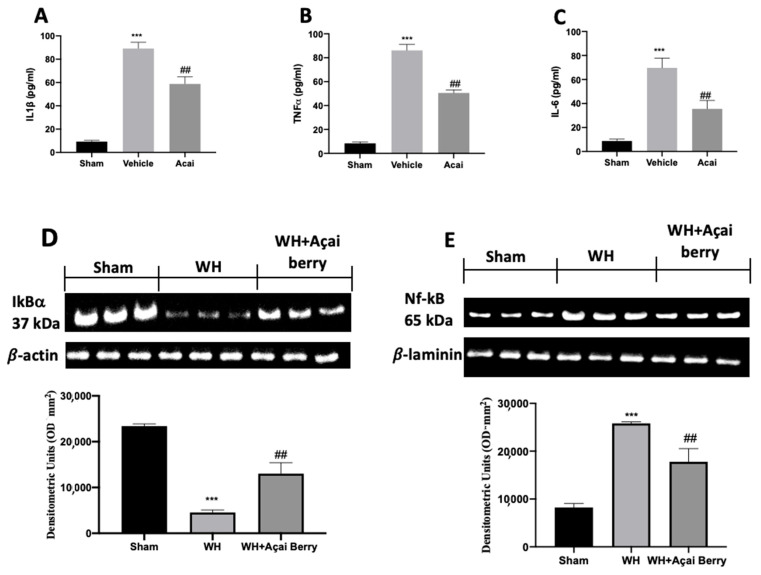
Expression of inflammatory cytokines: IL-1β (**A**), TNF-α (**B**), and IL-6 (**C**). Western blot analysis for IκB-α (**D**) and NF-κB (**E**). A demonstrative blot of lysates with a densitometric analysis for all animals is shown. Data are expressed as the mean ± SEM of N = 6 mice/group. *** *p* < 0.001 vs. sham; ## *p* < 0.01 vs. WH.

**Figure 7 ijms-24-00834-f007:**
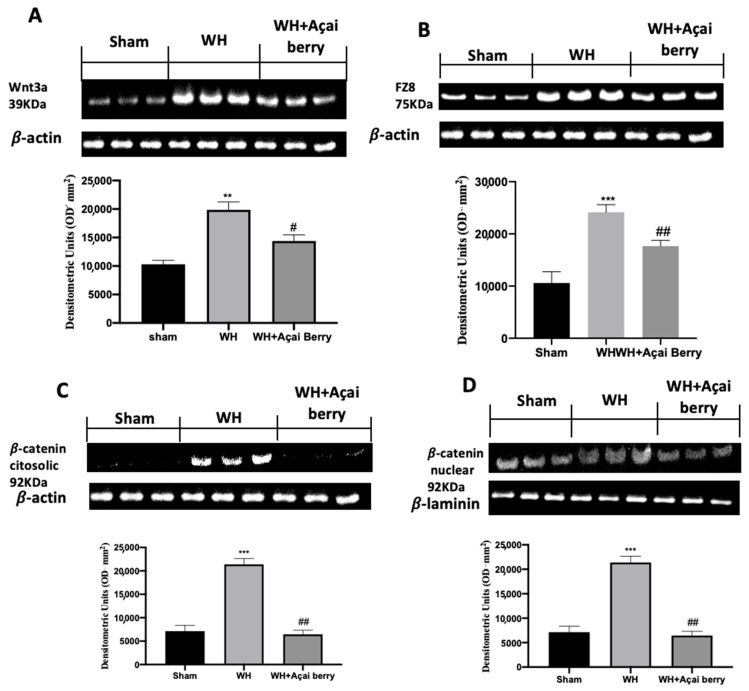
Western blot analysis of Wnt3a (**A**), FZ8 (**B**), cytosolic β-catenin (**C**), and nuclear β-catenin (**D**). A demonstrative blot of lysates with a densitometric analysis for all animals is shown. Data are expressed as the mean ± SEM of N = 6 mice/group. ** *p* < 0.01 vs. sham, *** *p* < 0.001 vs. sham; # *p* < 0.05 vs. WH; ## *p* < 0.01 vs. WH.

## Data Availability

The data used to support the findings of this study are available from the corresponding author upon request.

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
