# Peer review of "Açai Berry Administration Promotes Wound Healing through Wnt/β-Catenin Pathway"

_ijms, 2023, doi:10.3390/ijms24010834_

Round 1

Reviewer 1 Report

Dear authors

This is an interesting research related to evaluate  the wound healing efficiency of Açai berry. 

1. Please check all type errors and language throughout the manuscript. For example "4. Materials e methods"

2. Section "4. Materials and Methods" should be change to Section 2, it is better for readers can approach and undersatnd the next contents in Results and discussions.

3. In the section Materials and Methods, The authors didn't describe how to prepare the Açai berry and how to administrate it in/on mouses

4. All figures in this manuscript is inserted in a small scale, which is difficult for readers to see well. Please enlarge them in an appropriate one.

5. The results and discussions were separated and they didn't support each other well. I suggest the authors integrate them into one section "Results and Discussions"

6. The conclusions is very poor. It didn't present the significant results of this research. It should be revised and extended

Author Response

Author’s reply to Reviewer 1 

We thank the referee for the comments and replied to them point by point:

Please check all type errors and language throughout the manuscript. For example "4. Materials e methods"

We thank the referee for the comment and as he/her suggest corrected errors.

Section "4. Materials and Methods" should be change to Section 2, it is better for readers can approach and undersatnd the next contents in Results and discussions.

We thank the reviewer for the comment. Unfortunately, we can not change the template of the journal. However, we tried to make more clear the results and discussion sections to make the manuscript more easy to read.

In the section Materials and Methods, The authors didn't describe how to prepare the Açai berry and how to administrate it in/on mouses

We thank the reviewer for the comment and added the required information.

Freeze-dried Açai extract was dissolved in distilled water and administered orally at the dose of 500mg/Kg

 All figures in this manuscript is inserted in a small scale, which is difficult for readers to see well. Please enlarge them in an appropriate one.

We thank the reviewer for the comment. We have edited the images.

 The results and discussions were separated and they didn't support each other well. I suggest the authors integrate them into one section "Results and Discussions"

We thank the reviewer for the comment. Unfortunately, we can not change the template of the journal. However, we tried to make more clear the results and discussion sections to make the manuscript more easy to read.

 The conclusions is very poor. It didn't present the significant results of this research. It should be revised and extended

We thank the reviewer for the comment. We rewritten the paragraph better.

Here, we demonstrate that Açai berry was able to modulate activation of the WNT/b-catenin signaling pathway in a mouse model of wound healing. In particular, the oral administration of Açai berry was able to improve tissue damage, reduced cell infiltration and improved tissue quality. Additionally, it reduced the levels of TNF‐α and IL‐18 that strictly downstream of the WNT/β‐catenin pathway and modulates the activity of growth factors, as TGF- β and VEGF, which are the basis of the wound healing process. Açai berry also showed important anti‐inflammatory and antioxidant activities, reducing the activation of the NFkB pathway and Nitrotyrosine and PARP-1 expression

Additionally, the authors performed the English Editing and uploaded the certificate in the unpublished materials section. 

We strongly believe that the quality of our manuscript has been substantially improved and that now it is suitable for publication in IJMS.

Reviewer 2 Report

The authors investigate the effect of Acai berry administration on wound healing. The following comments and suggestions may improve the article quality.

- All figures are too small in size, I can't see any details, please enlarge it and enhance its resolution.

- Figure 1, the authors did not mention D plot in the figure title.

- Give brief comment of histopathological injury score and discuss your results of Figure 1D regarding that (line 97).

-

- The authors used freeze dried acai extract (500 mg/kg) in saline, please mention its solubility data, I know you select your doses based on previous study, but it is an important point.

- Mention the number of animals in each group.

- Data of Sham+ Acai group is not shown, that's Ok in figure 1D, but we need the H&D staining figure to confirm your findings.

- In the experimental and results sections, mention the number of animals in each group.

- Enhance the discussion part, correlate your results with discussion and dose dependent response, western plot discussion was not mentioned, only results, please give your insights.

- Conclusion needs improvement.

Author Response

                                                   Author’s reply to Reviewer 2

We thank the referee for the comments and replied to them point by point:

 All figures are too small in size, I can't see any details, please enlarge it and enhance its resolution.

We thank the referee for the comment and as he/her suggest corrected errors.

 Figure 1, the authors did not mention D plot in the figure title.

We thank the referee for the comment and as he/her suggest corrected errors.

Give brief comment of histopathological injury score and discuss your results of Figure 1D regarding that (line 97).

We thank the reviewer for the comment. we rewritten the paragraph better.

Oral administration of Açai berry showed an improvement in wound closure, granulation, and re-epithelialization, all of which contributed to a nearly complete restoration of the epidermis in this group.

The authors used freeze dried acai extract (500 mg/kg) in saline, please mention its solubility data, I know you select your doses based on previous study, but it is an important point.

We thank the reviewer for the comment and added the required information. Freeze-dried Açai extract was dissolved in saline, however, previous studies showed that it could be also dissolved in water. It was administered orally at the dose of 500mg/Kg. The dose and the route of administration of açai berry were chosen based on our previous study, with a dose-response experiment.

Impellizzeri, D.; D’Amico, R.; Fusco, R.; Genovese, T.; Peritore, A.F.; Gugliandolo, E.; Crupi, R.; Interdonato, L.; Di Paola, D.; Di Paola, R. Acai Berry Mitigates Vascular Dementia-Induced Neuropathological Alterations Modulating Nrf-2/Beclin1 Pathways. Cells 2022, 11, 2616.

Genovese, T.; D'Amico, R.; Fusco, R.; Impellizzeri, D.; Peritore, A.F.; Crupi, R.; Interdonato, L.; Gugliandolo, E.; Cuzzocrea, S.; Paola, R.D.; et al. Acai (Euterpe Oleraceae Mart.) Seeds Regulate NF-kappaB and Nrf2/ARE Pathways Protecting Lung against Acute and Chronic Inflammation. Cell Physiol Biochem 2022, 56, 1-20, doi:10.33594/000000529.

Mention the number of animals in each group.

We thank the reviewer for the comment and added the required information.

Data of Sham+ Acai group is not shown, that's Ok in figure 1D, but we need the H&D staining figure to confirm your findings.

We thank the reviewer for the comment and added the required information.

In the experimental and results sections, mention the number of animals in each group.

We thank the reviewer for the comment and added the required information.

The animals were randomly distributed into the following group (n=12):

Enhance the discussion part, correlate your results with discussion and dose dependent response, western plot discussion was not mentioned, only results, please give your insights.

We thank the reviewer for the comment and we rewritten the paragraph better.

Conclusion needs improvement.

We thank the reviewer for the comment and we rewritten the paragraph better.

Additionally, the authors performed the English Editing and uploaded the certificate in the unpublished materials section. 

We strongly believe that the quality of our manuscript has been substantially improved and that now it is suitable for publication in IJMS.

Round 2

Reviewer 1 Report

I think it can be accepted for publication

Reviewer 2 Report

Thanks for the authors for their commitment and enhancement of the manuscript quality.